# The Acceptance of Cream Soups with the Addition of Edible Insects (Mealworm, *T. molitor*; House Cricket, *A. domesticus*; Buffalo Worm, *A. diaperinus*; Grasshopper, *R. differens*) among Young People and Seniors in Poland

**DOI:** 10.3390/nu15245047

**Published:** 2023-12-08

**Authors:** Magdalena Skotnicka, Aleksandra Mazurek, Stanisław Kowalski

**Affiliations:** 1Department of Commodity Science, Medical University of Gdańsk, 80-211 Gdańsk, Poland; aleksandra.mazurek@gumed.edu.pl; 2Department of Carbohydrate Technology and Cereal Processing, Faculty of Food Technology, University of Agriculture in Krakow, 31-120 Kraków, Poland; rrkowals@cyf-kr.edu.pl

**Keywords:** edible insects, mealworm, buffalo worm, house cricket, grasshopper, soup, consumer acceptance

## Abstract

Research on the acceptance of consuming insects in one’s diet shows the increasing importance of this issue in the context of a sustainable food chain and ecology. Insects represent a promising food source due to their high nutritional value, efficiency in production, and minimal environmental impact, as well as the growing awareness of ecological issues. Despite these benefits, cultural and psychological barriers hinder the acceptance of consuming insects in Western countries. In this study, an assessment was made of the acceptance level of cream-type soups made from tomatoes and white vegetables with the addition of 20% flour from four insect species: mealworm (*T. molitor)*; house cricket (*A. domesticus*); buffalo worm (*A. diaperinus*); and grasshopper (*R. differens*), compared to a control sample. One hundred and four subjects (55 seniors and 49 young adults) participated in this study. The acceptance level of various soups with insect flour was evaluated, considering different sensory parameters such as appearance, smell, taste, and texture. The research showed that older people have a lower acceptance for dishes containing insects compared to young adults, but the differences in the responses given were not statistically significant (*p* = 0.05), rejecting the assumption that insect-based products should be mainly targeted at young people. Of all the proposed test samples, the mealworm (*T. molitor*) was the most acceptable insect species in the tests in both taste versions for both age groups. The average score was 6.63 points on a 10-point scale. The products with the addition of grasshopper (*R. differens*) were rated the lowest. The acceptance level ranged between 4.23 and 4.38 points. A multiple regression analysis showed that taste and texture had the strongest influence on the overall acceptance of these dishes, and the results obtained were highly correlated with the general opinion of the testers. The increasing acceptance level and growing interest in this type of food can be a positive step towards sustainable and efficient food production.

## 1. Introduction

Insects have been known as a food source for various population groups for centuries, but only recent years have drawn attention to them as highly nutritious products, inexpensive in production, safe, and tasty [1,2,3,4,5]. In some parts of the world, the introduction of insects into the diet is natural and does not meet resistance, while incorporating them into the diets of Europeans or North Americans is quite challenging. Until recently, insects in Western countries were produced only for feed markets. Currently, an increasing number of countries are breeding, producing, processing, and selling edible insects in various forms commercially. Many factors influence the development of the insect market: better-adapted and more consistent legal regulations [6,7], increasing production safety and euthanasia methods, and a broader product range for individual consumers. An essential factor is also the greater consumer awareness and improved education about the benefits of consuming insects [8,9,10]. Undoubtedly, one of the most critical premises is operating based on a sustainable development strategy, where the benefits of insect production in every respect surpass conventional livestock farming. The increasing demand for various kinds of food has become an essential element of building food security. All of these determinants and challenges posed to the food industry mean that interest in the edible insect market is growing. It is estimated that the share of aquafeed will dynamically increase from 17% to 40%, including edible insects [11]. Both Rabo Bank and IPIFF data suggest an upward market trend in the coming years [12]. There are about 1600–2000 insect species worldwide sourced from nature and used in daily diets [13]. Edible insects vary considerably. Their nutritional value depends on many components, such as the insect species, breeding substrate, breeding conditions, and processing method. Insects are a valuable source of protein compared to traditional meat products. They contain essential vitamins and minerals [14,15,16], especially being rich in iron and zinc [17,18]. Apart from basic nutrients, insects also contain several bioactive compounds and chitin, which makes them even more valuable as food in diets [19]. Insects differ in taste, shape, use, maturity stage, and serving form. The most significant barrier in Western countries is the lack of taste acceptance and a high level of neophobia, often resulting from prejudices and a lack of knowledge [20,21,22]. On the other hand, these countries have a strong awareness of taking care of the planet and the environment, so it is suggested that acting for higher goals might be an impetus to introduce insects into daily diets. From a consumer’s perspective, the most crucial element is full and complete information about the origin, safety, and durability of insects. The serving method is also vital. Insects are most willingly consumed in the form of a powder, added to commonly known and liked products [23,24]. The market offer is becoming richer every year. Insect flour or partly crushed insects can be an addition to cereal products (bread, biscuits, cookies, muesli, pasta) [25,26,27,28,29], sweets (chips, bars, lollipops) [30,31], as substitutes or meat additions (cutlets, burgers) [23], and as additions to salads, dips, and sauces. Most studies on acceptability indicate a significant market potential for various insect-based products, even in highly developed countries. The vast majority of recent publications suggest that consumer acceptance ratings for such products are relatively high [32,33,34,35]. The possibilities seem endless. From the perspective of palatability and acceptance level, adding insects to soups can be an interesting proposition. They can be added in the form of a powder or as whole insects, either larvae or imago. Soups are a product willingly consumed worldwide, but there are few studies on using insects in soups. Some studies were conducted in African and Asian countries, but on a very limited scale [36,37,38,39]. However, it seems that soups are one of the best basic products that can be enriched with insect-based products. Many authors suggest that the target group for insect-based products should be the younger generations, since the already-established dietary patterns of older people do not allow for such drastic changes [40,41,42]. Whether this is entirely true was something investigated in the authors’ studies. Considering these premises, the acceptance level of cream-type tomato soups and white vegetable soups with the addition of various insect variants was examined in two extreme age groups of young adults and seniors.

## 2. Materials and Methods

### 2.1. Materials

Flours from the following freeze-dried edible insects were used in this study: mealworms (*Tenebrio molitor*; TM), buffalo worms (*Alphitobius diaperinus;* BW), house crickets (*Acheta domesticus*; CR), and grasshoppers (*Ruspolia differens*; GH). The freeze-dried insects were acquired from a breeding facility in the Netherlands (Insecten kwekrij van de Ven Fortweg, Deurne, The Netherlands). The first two were in the larval form, and the other two were in the imago form. The insects were then ground in a laboratory mill (IKA, A11 basic, Staufen, Germany), and the obtained flour was passed through a sieve.

### 2.2. Preparation of Soup with Insect Flour

For this study, two types of cream soups were prepared: tomato and white vegetable. For the tomato soup, fresh tomatoes were used. After peeling and adding onions, they were sautéed in a pan with rapeseed oil. A broth was then made following a traditional recipe and was added to the vegetables. The soup was cooked for 30 min. Finally, the soup was blended using a kitchen blender.

A similar procedure was followed for the white vegetable soup. Vegetables (onion, leek, parsley, celery, potatoes) were diced, sautéed in a pan, and then covered with broth. As with the first soup, it was cooked for 30 min, after which 30% cream was added. It was then blended into a smooth cream.

The caloric value was calculated per 100 mL of soup. The tomato soup had a value of 85 kcal/100 mL, and that of the white vegetable cream soup was 114 kcal/100 mL. In this manner, the base dishes were prepared as control samples. The caloric value was calculated using the “Dieta 6.0” diet planning and energy value counting software, developed by the National Institute of Public Health, based on the Polish food composition database.

The next stage of the study involved adding 20% flour from four species of edible insects relative to the dry weight of the control sample. Both soup recipes were standardized such that the 20% addition always equated to 5 g of flour/100 g of soup. The 20% addition of insect flour was based on previous analyses, where a lower content of insect flour was near the threshold of detectability. Too high a content significantly altered the taste and texture, resulting in samples being completely rejected by consumers.

Finally, ten soups were prepared for testing: five tomato soups (C—control; TCMW—with mealworm; TCBW—with buffalo worm; TCCR—with cricket; and TCGH—with grasshopper) and five white vegetable soups (C—control; WVMW—with mealworm; WVBW—with buffalo worm; WVCR—with cricket; and WVGH—with grasshopper). The soups were served immediately after preparation in white disposable bowls. Each serving contained 50 mL of soup and was served at a temperature of 70 °C in accordance with collective catering guidelines [43].

### 2.3. Chemical Composition

The following parameters of the edible insect flour were analyzed: the ash content (AOAC 923.03); protein content (AOAC 950.36) (the protein content was calculated applying a conversion factor of 6.25); crude fat content (AOAC 935.38); water content (AOAC 925.10); and the total, soluble, and insoluble dietary fiber content (AOAC 991.43) [44]. The determination of the chitin content was adapted based on reference [45]. The analyses were performed in triplicate.

### 2.4. Characteristics of the Study Group

In the first step, 165 individuals (from two age groups) from the volunteer database of the Department of Clinical Nutrition at the Medical University of Gdańsk were invited to participate in the study. Out of these, 141 individuals ultimately qualified for the study who did not exhibit food neophobia (FNS) towards consuming insects. These 141 individuals were divided into two distinct age groups: young adults aged 18–29 years (Me = 25.6) and seniors above 65 years old (Me = 72.3). The study was completed by 49 young consumers and 55 seniors, resulting in a total of 104 participants, as presented in Figure 1. Twenty-one seniors and 16 young adults did not complete the study. Four seniors and 2 individuals from the younger group were ill or hospitalized, while the rest refused to continue the study or did not show up for the study without giving a reason.

### 2.5. Consumer Acceptance Analysis

A sensory analysis of the prepared two types of soups (tomato and white vegetable) was conducted by 104 panelists divided into two age groups. The evaluators responded on pre-prepared rating cards. Each evaluator received their own set of samples (4 samples) in a random order of one type of soup, and after two days, a set of four samples of the second soup. The soups were served and consumed warm.

A 10 cm unstructured scale was used for the evaluation, with appropriate markings for the specific descriptors at both ends. The following descriptors were adopted: appearance (0 (totally unsatisfactory)–10 (totally satisfactory)), taste (0 (very unpalatable)–10 (very tasty)), texture (0 (totally unsatisfactory to me)–10 (totally satisfactory to me)), and acceptance concerning the species and variant of the insect (0 (very unsatisfactory)–10 (very satisfactory)). The choice of descriptors considered in the sensory evaluation was based on PN-EN ISO 5492:2009 [46] and PN-ISO 11035:1999 [47].

Names and definitions of descriptors:

*Appearance*: The overall visual impression that the product evokes, composed of a series of individual visually perceptible features (e.g., shape, color, shine).

*Smell*: The sensation perceived by the sense of smell.

*Taste*: The sensation perceived by the taste receptors on the tongue’s surface when stimulated by a stimulus.

*Texture*: The sound intensity associated with deforming a sample (e.g., when biting into it).

*Acceptance*: Overall sensory feelings of the consumer towards the presented sample: their acceptance or rejection conditioned by the product’s quality or consumers’ living standard. For acceptance evaluation, a hedonic scale was used in the form of a linear scale, with appropriate edge labels “very unsatisfactory-very satisfactory”. The evaluators were healthy and did not take medications, supplements, or special diets. All participants voluntarily signed a research consent form approved by the Independent Bioethical Committee for Scientific Research at the Medical University of Gdańsk (NKBBN/346/2021). The study complies with the ethical principles of non-maleficence, beneficence, justice, and autonomy contained in the ethical provisions of the Helsinki Declaration of 1975, revised in 2013. Before the study, everyone completed the food neophobia study (FNS) questionnaire [48]. Only those individuals who did not exhibit food neophobia participated in the study.

### 2.6. Statistical Analysis

The statistical analysis was performed using the PQStat Software (2023) v.1.8.6.102. Initially, the distribution of all variables was checked to determine whether they involved a normal distribution or not and if non-parametric tests were necessary. The normality of data distribution was assessed using the Shapiro–Wilk test. For a comparison of two independent groups, the Mann–Whitney U test was applied. The Wilcoxon signed-rank test was used to rank the significance of individual quality indicators in relation to the control sample.

It was assumed that the appearance, smell, taste, and texture influence the acceptance of edible insects (both in terms of flavor variants and species), which can be represented as a functional relationship: f (insect acceptance) = f (insect appearance, insect smell, insect taste, insect texture). To determine the importance of individual components, it was necessary to identify the values of the influence coefficients of individual components on the acceptance value. Therefore, the above function took the form of a model in a mathematical notation:Acceptance = x_1_·appearance + x_2_·smell + x_3_·taste + x_4_·texture

x_1_—coefficient of appearance;

x_2_—coefficient of smell;

x_3_—coefficient of taste;

x_4_—coefficient of texture.

## 3. Results

The basic chemical composition of the flour from the selected edible insects was first examined, as presented in Table 1. The highest total protein content was found in *A. domesticus*. The fat content was also the highest for this insect compared to other analyzed samples. The fat content of the grasshoppers was three times less than that of the house crickets. In all cases, trace amounts of water-soluble fiber were observed. On the other hand, the flour from field crickets and house crickets was the richest in insoluble fiber and chitin, which is likely related to the form of the insect (imago) from which these two flours are derived.

In the analysis, four soups with a 20% addition of insect flour (MW—mealworm, BW—buffalo worm, CR—house cricket, GH—grasshopper) were used compared to the control sample. The concept involved a consumer evaluation of the soups in two flavor variants: (1) tomato cream soup and (2) cream soup from white vegetables for each variant. Two age-extreme groups participated in this study: young adults and seniors over 60. This study utilized two extreme age groups to indicate potential differences between the groups in the acceptance evaluation of insect-based products. The youngest adults are perceived as the most willing and open to trying new products and flavors. Meanwhile, individuals over the age of 65 are considered the most conservative group, reluctant to change dietary habits and try new items in the food market.

Before starting this study, testers were asked to complete the neophobia scale (FNS). The majority of respondents, 58.8%, belonged to the neutral group, while neophiles accounted for 21.5%. These results draw attention to the fact that respondents simultaneously have a fear of the new and seek novelty. The high standard deviation values and variability coefficient also indicate this, with the percentage of neutral respondents being higher among seniors, accounting for 62.3% compared to 54.3% for young adults. Ultimately, those from the neutral group and the neophile group were qualified for the next stage of the study.

This study assumed that the acceptance results and individual quality distinctions would differ depending on the age of the respondents. To compare the two groups, the similarity of the distribution of variables to a normal distribution was checked using the Shapiro–Wilk test. In the case of the current data, all nominal variables are not normal distributions, so non-parametric tests were used for them at *p* = 0.05. The acceptability assessment results for each age group, young (Y) and seniors (S), within the basic sensory distinctions, are presented in Table 2. To demonstrate statistically significant differences between two independent groups, the Mann–Whitney U test was used.

Generally, seniors rated all the sensory attributes of the insect-based soups lower, including the control samples. The control sample was rated highest in both age groups and in both flavor variants for the tomato and cream of white vegetable soups. However, it is worth noting that the control sample (STC) was rated slightly higher than SWV. This was reflected in the acceptance ratings of soups with insect additives. The tomato soup consistently achieved higher acceptance levels. The more pronounced taste, color, and smell likely offset the impact of the insect addition. However, the differences in ratings between the different variants were not statistically significant, which means that no significant relationship was found between acceptability depending on the age of the respondents. Detailed data are shown in Table 3; therefore, a subsequent analysis was carried out without dividing by respondent age. For individual sensory attributes (appearance, taste, smell, texture), no statistically significant differences were found.

The initial rating for the tomato soup (STC) was nine points on a 10-point scale and 8.5 points for the cream of white vegetable soup (SWV). The appearance of the soups was rated highest for both control samples (STC, SWV). Comparing the *p*-value (one-tailed) = 0.00005 of the Wilcoxon test based on the T-statistic with the significance level α = 0.05, it was determined that there was a statistically significant difference in the appearance rating between the control sample and the appearance rating in other tested variants. The appearance ratings of soups with the addition of the four insects were consistently lower than those of the similarly rated tomato soup in recipes that did not use that insect (the sum of the negative ranks is much greater than the sum of the positive ranks). The same conclusion is suggested by the Wilcoxon test based on the Z statistic with a one-tailed *p*-value ≤ 0.000001. The appearance was rated lowest for the grasshoppers (TCGH) and crickets (TCCR), as illustrated in Figure 2a.

An analogous statistical analysis was conducted for the cream of white vegetable soup. In this variant, the appearance of soups with the addition of insects was also significantly lower, with *p* < 0.005, as depicted in Figure 2b. The appearance of the soup with the addition of grasshoppers (WVGH = 4.85) was rated the lowest, and this rating was even lower than that for the tomato soup based on grasshoppers (TCGH). In terms of aroma evaluation, a decline in acceptance was noted for all test samples. In all instances, the decrease in aroma acceptance for soups with added insects was statistically significant at a *p*-value (one-sided) of 0.00005 based on the Wilcoxon T-statistic test with a significance level α = 0.05. The sum of the negative ranks was much greater than the sum of the positive ranks, which is evident in Figure 3a,b.

The key determinant influencing the overall acceptance level was the taste. The taste in the control sample was rated at 8.31 for STC and 8.41 for SWV on a 10-point scale for the tomato soups and the white vegetable soups. For the tomato soup with the addition of mealworm (TCMW) and buffalo worm (TCBW), there was a statistically significant decrease in the taste rating, with an average decrease in taste acceptance by two points. The lowest rating was given to the soups with the addition of grasshopper, where in the case of TCGH the average rating was 4.22. Even lower ratings were given to the white vegetable soups with the addition of grasshopper (WVGH), where each taste acceptance rating was lower than four, essentially disqualifying this soup variant in terms of taste. This is clearly presented in Figure 4a,b.

Among all the features affecting acceptance, the texture, or rather the consistency of soups, is extremely important. This is especially a key parameter for cream soups. In the current study, it is visible in Figure 5a,b that the texture rating was dependent on the type and form of the insect from which the flour was made. Creams based on insects in the imago form (TCCR, TCGH, and WVCR, WVGH) were rated significantly lower than the samples with mealworms and buffalo worms in the larval form. It was found that there is a statistical difference in the level of texture acceptance between the control sample and all soups with the addition of insect flour. Each time, comparing the *p*-value (one-sided) = 0.00005 of the Wilcoxon test based on the T-statistic with the significance level α = 0.05 showed this dependency. The sum of negative ranks is much greater than the sum of positive ranks. The same decision would also be made based on the one-sided *p*-value = < 0.000001 of the Wilcoxon test based on the Z-statistic.

In the next stage, it was verified whether all the considered sensory descriptors (appearance, scent, taste, texture) were related to the acceptance level of the analyzed soups. Figure 6a,b depict a graphical representation of the research results. The analysis of the average scores for the overall quality showed that the soups without the addition of insect flour were characterized by the highest acceptance level. Among the samples with the addition of insects, the tomato soup based on mealworm (TCMW) had the highest acceptability determined by the sum of all descriptors. The addition of mealworm flour to the white vegetable soup was rated slightly lower, similar to the white vegetable cream soup with the addition of the buffalo worm. The lowest acceptance level was achieved by soups enriched with flour from field crickets. For both TCGH and WVGH, the overall acceptance level was expressed in values of 4.35 and 4.28 points, respectively.

The results of the sensory analysis became the basis for estimating the relationship between the acceptance of the selected sensory quality descriptors of soups with the addition of insect flour and the acceptance of these products, perceived through the lens of their overall quality assessment. For this purpose, a multiple regression analysis was applied, which allowed for the development of acceptance models. The dependent variable was the acceptance level of the soups, while the independent input variables were the acceptance ratings of individual sensory quality parameters: appearance, scent, taste, and texture. During the analytical procedure, independent variables were narrowed down to critical parameters. The significance of the generated acceptance models was assessed at a significance level of *p* ≤ 0.05. A multiple regression analysis was used for all the soup variants studied to estimate the influence of individual variables on the overall acceptance level. Table 4 showcases the obtained regression equations for the overall acceptability.

The multiple regression analysis showed that in the case of the tested soups with insect flour, the aroma was insignificant and had no influence on the overall acceptability of the tested soups, or its role was marginal. Only in the case of the white vegetable soup with the addition of grasshopper flour did the aroma play a significant role. In this case, the texture did not significantly impact the acceptability, unlike in other tested variants. In summary, the obtained results indicate that the level of acceptability resulted from predictors included in the regression equation, with taste, appearance, and texture having the greatest importance for overall acceptability and aroma playing a smaller role. The coefficient of determination (R2) for all tested samples was high, ranging from 0.97 to 0.99, which means that the generated model explains almost 97–99% of the variability of the dependent variable. Only 1–3% of the variability was determined by other unanalyzed parameters. The regression analysis showed that the taste and texture were critical determinants of the overall acceptability. In cases where the overall acceptability rating was the lowest, the taste and texture had the highest importance. High coefficients for x3 and x4 made taste and texture the most significant factors influencing the acceptability and final evaluation of the participants in the study.

## 4. Discussion

The development of entomophagy, especially in Western countries, faces many challenges. Products based on insects or containing them as ingredients are timidly entering the market. However, the food industry continues to grapple with negative perceptions. The level of neophobia towards new foods remains consistently high in Europe and highly developed countries [22,49,50,51,52]. However, preliminary findings indicate that the aversion level might not be as high as suggested by the mentioned authors. The current research is focused on barriers and factors influencing consumers’ willingness to consume insects and methods of presenting them.

Seniors, who are considered the most conservative age group when it comes to introducing new products into their diets, participated in this study [20,53,54,55]. In the case of the current experiment, older individuals displayed great courage and willingness to participate, especially since it involved well-liked soups in two flavors.

A review of the available literature indicates that insects have served as the foundation for many functional products such as snacks, sweets, bread, dairy, and meat [23,25,56,57]. However, there have not been extensive studies on the acceptance of insects in soups, especially among European consumers. Soups are a popular, common, and delicious food product recommended for all age groups and populations. Technologically, it is relatively easy to formulate soups with insect flour, especially since insect flours have excellent rheological properties [58,59,60]. Soups with insect additives can also be successfully used in the form of freeze-dried soups and powders, making them convenient and functional foods [61,62]. Introducing insects into soups might be easier for some consumers to accept, especially if the insects are finely ground or in a powder form, as in our study.

Adeboye et al. proposed enriching vegetable soups with termite flour and used several variants of soups with different proportions of termite flour made from *Macrotermes bellicosus*. All samples were positively accepted by the evaluators, with no significant differences in taste, texture, consistency, or appearance. The soup with a 10% addition of termite flour received the highest rating [36]. Similar studies were conducted by Thomas and Olatunji, who enriched traditional soup and egusi soup with *Cirina forda* larvae. The nutritional content of the insect-enriched soups was higher and correlated with higher acceptability than traditional portions [39]. However, both studies were conducted among African populations, where insects are common in their diets. Therefore, it is essential to conduct studies among consumers in countries where edible insect consumption is still relatively low and unpopular.

The analysis of the samples used in this study suggests that the control samples without insect flour received the highest ratings. Overall, all test samples based on tomato soup were rated significantly higher than the cream of white vegetable soup, considering both the overall acceptability and individual descriptors. The higher rating for tomato soup was likely due to the influence of a more pronounced taste and intense color, potentially making it easier to “hide” the characteristic taste and appearance imparted by the insect additive [63]. Similar studies have been conducted on other food products, where the color and rheological characteristics were determining factors that effectively lowered overall sensory scores [52,64]. On the other hand, Meyer-Rochow and Hakko conducted an experiment to assess the level of identification of insect-based food by blindfolded volunteers. It was found that recognition was low, which may be evidence that adding powdered insects to food does not necessarily alter the taste or consistency [65].

As mentioned earlier, the control samples without insect flour received the highest ratings, while the highest overall acceptability among soups with a 20% insect flour addition was obtained by tomato soup with the addition of mealworm flour. This is not surprising, as many studies have indicated that mealworms typically achieve the highest acceptance, especially in terms of taste, often described as slightly nutty. The current findings align with previous research by other authors, who pointed out that mealworms are a species of insect that receives the highest acceptance and is most commonly consumed [32,64,66].

Additionally, *T. molitor* is used in human nutrition in the form of larvae, which are bright yellow without a dark chitinous exoskeleton, which may affect their acceptability. This would also confirm the fact that the second-highest acceptability rating was obtained by soups with the addition of buffalo worm, which was also in the larval form. This had particular significance in terms of the appearance and texture evaluation. Perhaps this is a hint to producers that insects in their larval form are characterized by higher acceptability, provided that they are finely ground [67]. This is supported by the research of Mazurek et al., who assessed the acceptability of pancakes with the addition of various insects in different stages of development. The analysis suggested that the acceptability rating for products, in this case, pancakes with insect larvae flour, is higher in terms of taste, appearance, and structure [68]. The grasshopper received the lowest rating in both soups and by both groups of participants. Previous studies by the authors also confirmed a low level of acceptability of products with the addition of *R. differens* [69,70], although research by Cruz-Lopez [71] on the acceptability of sausages with grasshopper additives showed a positive response from researchers. Kim et al. evaluated the sensory properties of *Oxya chinensis sinuosa* (grasshopper) in enriched yogurts. It was found that the taste and aroma were the characteristics that effectively lowered the overall rating, not the texture and color, similar to the results of the current study. Perhaps the base product and the appropriate recipe to which the insect powder is added are essential factors to consider [66].

Considering the assessment of acceptance, it is important to understand which descriptors are decisive in the overall evaluation. This is especially significant when introducing new products to the market. With this information, products can be better designed based on appropriate marketing strategies. In the case of the current study, based on a multiple regression analysis, the taste and texture of soups with a 20% insect flour addition were identified as the most important determinants shaping their overall acceptability. This shows that when proposing insect-based food to consumers, it must meet the criteria of palatability. Therefore, extensive sensory studies on various insect-based food products in different forms are essential for effectively designing food with high sensory qualities that consumers, even the most conservative ones, will accept.

A notable advantage of this study is that, for the first time, the level of acceptability of soups with the addition of four different insects was assessed in two extreme age groups and two different flavor variations. The research results allowed for the determination of acceptability levels for individual insects, divided into two age groups. The small differences between these age groups suggest that acceptability would be at a similar level in other age groups. The current findings also demonstrate that the key element is not only the type of added insect but also the base product and a properly selected recipe.

Although this study contributes valuable, organized information and guidance for food producers to the scientific literature, it has some limitations. Many studies had to be excluded due to not meeting previously specified inclusion criteria. Not all information regarding motivation, attitudes, behaviors, and factors determining the consumption of edible insects was collected, which could expand knowledge regarding insect consumption. Furthermore, the inclusion of edible insects and derivative products in the human diet can increase dietary diversity. Besides vegetarian diets, it can promote proper nutrition in individuals with high dietary requirements, such as athletes, pregnant women, the elderly, and undernourished individuals. Insects have significant potential as a food source due to their environmental and health benefits, but research on the commercialization of insect-based foods is still in its early stages. These results have high practical value directed towards dietary interventions and addressing environmental challenges, as well as meeting the modern requirements of all food market participants.

## 5. Conclusions

Some key findings from this work can contribute to expanding the existing knowledge about the factors influencing the acceptance of insects as food among different age groups. However, it has been demonstrated that the acceptability of food containing insects as hidden ingredients can be significantly increased, in contrast to consuming whole insects. Soups are a good base product for the production of insect-enriched food. Among the four species of insects studied, the highest level of acceptance, both in tomato soup and cream of white vegetables, was associated with mealworms (*Tenebrio molitor*)*,* while the product with the addition of grasshoppers (*Ruspolia differens*) received the lowest ratings. A multiple regression analysis indicated that taste and texture were the key descriptors shaping overall acceptability. The conducted research has shown that there is no statistically significant relationship between the age groups studied, which allows for expanding the target consumer group for insects. According to the earlier assumptions, the market offer should not be limited to the youngest generation. Based on these findings, future research should focus on marketing strategies and social interventions aimed at informing consumers, modifying behavior, educating, and creating new, attractive food products that incorporate edible insects.

## Figures and Tables

**Figure 1 nutrients-15-05047-f001:**
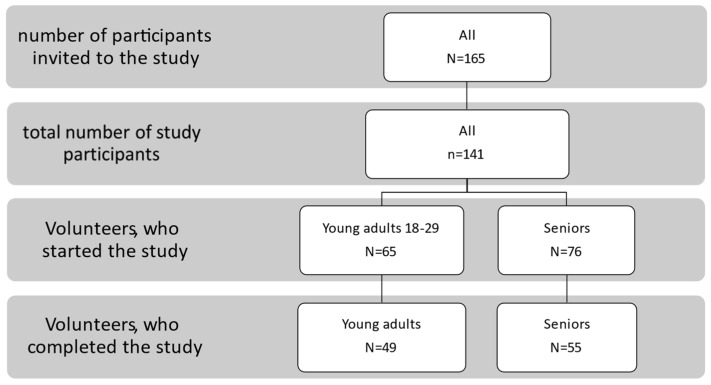
Description of the study group.

**Figure 2 nutrients-15-05047-f002:**
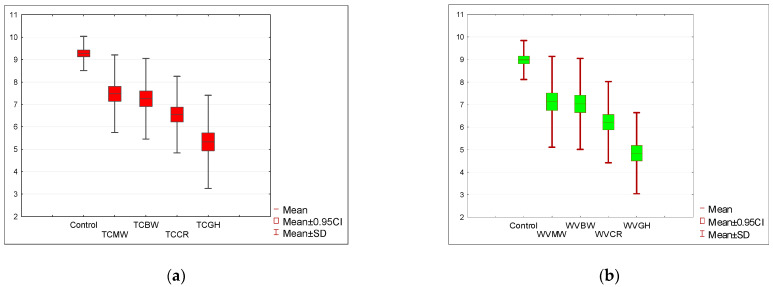
(**a**) Acceptance level of the appearance of tomato soup with the addition of insect flour compared to the control sample. (**b**) Acceptance level of the appearance of cream of white vegetable soup with the addition of insect flour compared to the control sample. Tomato soup: control, *T. molitor* (TCMW), *A. diaperinus* (TCBW), *A. domesticus* (TCCR), and *R. differens* (TCGH). White vegetable soup: control, *T. molitor* (WVMW), *A. diaperinus* (WVBW), *A. domesticus* (WVCR), and *R. differens* (WVGH).

**Figure 3 nutrients-15-05047-f003:**
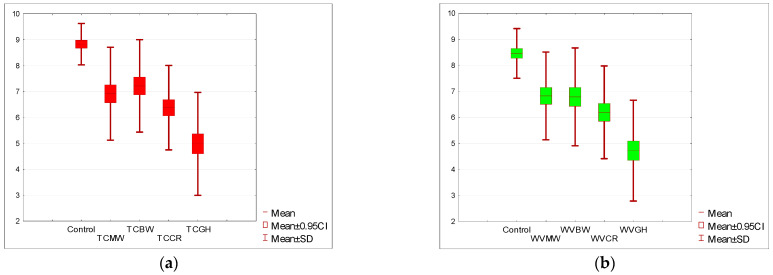
(**a**) Acceptance level of the aroma of tomato soup with added insect flour compared to the control sample. (**b**) Acceptance level of the aroma of cream of white vegetable soup with added insect flour compared to the control sample. Tomato soup: control, *T. molitor* (TCMW), *A. diaperinus* (TCBW), *A. domesticus* (TCCR), and *R. differens* (TCGH). White vegetable soup: control, *T. molitor* (WVMW), *A. diaperinus* (WVBW), *A. domesticus* (WVCR), and *R. differens* (WVGH).

**Figure 4 nutrients-15-05047-f004:**
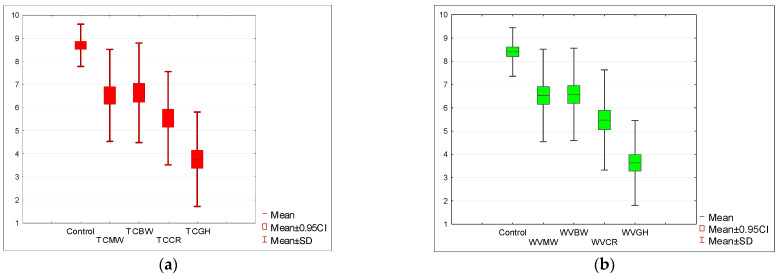
(**a**) Acceptance level of the taste of tomato soup with insect flour compared to the control sample. (**b**)**.** Acceptance level of the taste of cream of white vegetable soup with insect flour compared to the control sample. Tomato soup: control, *T. molitor* (TCMW), *A. diaperinus* (TCBW), *A. domesticus* (TCCR), and *R. differens* (TCGH). White vegetable soup: control, *T. molitor* (WVMW), *A. diaperinus* (WVBW), *A. domesticus* (WVCR), and *R. differens* (WVGH).

**Figure 5 nutrients-15-05047-f005:**
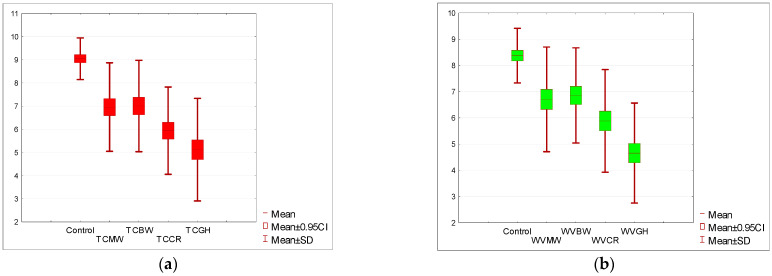
(**a**) Acceptance level of the texture of tomato soup with the addition of insect flour compared to the control sample. (**b**) Acceptance level of the texture of white vegetable soup with the addition of insect flour compared to the control sample. Tomato soup: control, *T. molitor* (TCMW), *A. diaperinus* (TCBW), *A. domesticus* (TCCR), and *R. differens* (TCGH). White vegetable soup: control, *T. molitor* (WVMW), *A. diaperinus* (WVBW), *A. domesticus* (WVCR), and *R. differens* (WVGH).

**Figure 6 nutrients-15-05047-f006:**
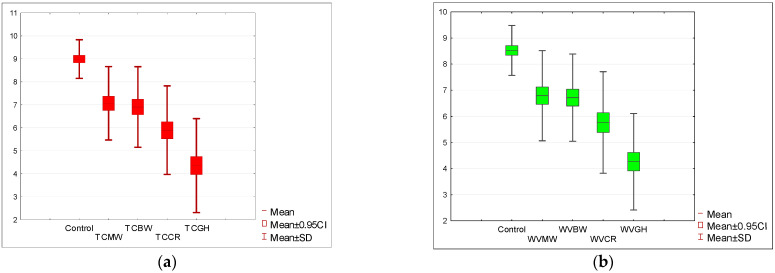
(**a**) Overall acceptance level of tomato soup with insect flour compared to the control sample. (**b**) Overall acceptance level of white vegetable soup with insect flour compared to the control sample. Tomato soup: control, *T. molitor* (TCMW), *A. diaperinus* (TCBW), *A. domesticus* (TCCR), and *R. differens* (TCGH). White vegetable soup: control, *T. molitor* (WVMW), *A. diaperinus* (WVBW), *A. domesticus* (WVCR), and *R. differens* (WVGH).

**Table 1 nutrients-15-05047-t001:** Chemical composition of flour-based edible insects (%).

Kind of Insect	Moisture	Protein	Ash	Fat	Dietary Fiber g × 100 g^−1^			
					Insoluble Fraction including Chitin	Chitin	Soluble Fraction	Total
*T. molitor*	1.56 ± 0.01 ^c^	45.39 ± 0.06 ^c^	3.86 ± 0.01 ^d^	14.29 ± 0.06 ^c^	11.56 ± 0.07 ^d^	6.93 ± 0.02 ^d^	0.36 ± 0.02 ^a^	11.92 ± 0.05 ^d^
*A. diaperinus*	5.40 ± 0.03 ^a^	49.51 ± 0.16 ^b^	4.71 ± 0.03 ^a^	26.44 ± 0.22 ^b^	12.96 ± 0.04 ^c^	7.33 ± 0.01 ^c^	0.00 ± 0.00	12.96 ± 0.04 ^c^
*A. domesticus*	2.47 ± 0.01 ^b^	55.18 ± 0.11 ^a^	4.34 ± 0.01 ^c^	29.01 ± 0.04 ^a^	18.48 ± 0.06 ^b^	9.92 ± 0.01 ^b^	0.24 ± 0.01 ^b^	18.72 ± 0.05 ^b^
*R. differens*	0.00 ± 0.00	42.96 ± 0.07 ^d^	4.48 ± 0.04 ^b^	9.60 ± 0.08 ^d^	19.94 ± 0.06 ^a^	10.83 ± 0.01 ^a^	0.00 ± 0.00	19.94 ± 0.06 ^a^

Values in the same row marked with different letters are statistically significantly different at *p* < 0.05 ± SD.

**Table 2 nutrients-15-05047-t002:** Mean acceptance values of sensory quality features of insect-based soups.

**Tomato Soup**										
	**Young**					**Seniors**				
	**Descriptors**									
**Kind of Edible Insects**	**Appearance**	**Odor**	**Taste**	**Structure**	**Acceptance**	**Appearance**	**Odor**	**Taste**	**Structure**	**Acceptance**
Control	9.38 ± 0.78 ^a^	8.90 ± 0.28 ^a^	8.28 ± 0.99 ^a^	9.06 ± 0.96 ^a^	9.04 ± 0.94 ^a^	9.18 ± 0.74 ^a^	8.75 ± 0.81 ^a^	8.48 ± 0.85^a^	9.03 ± 0.85 ^a^	8.95 ± 0.75 ^a^
TCMW	7.66 ± 1.79 ^b^	6.90 ± 1.84 ^c^	6.62 ± 2.04 ^b^	7.11 ± 1.83 ^b^	7.18 ± 1.52 ^b^	7.30 ± 1.68 ^b^	6.03 ± 1.77 ^c^	6.46 ± 1.96 ^b^	6.84 ± 1.97 ^b^	6.98 ± 1.66 ^b^
TCBW	7.34 ± 1.87 ^b^	7.28 ± 1.75 ^b^	6.78 ± 2.20 ^b^	6.93 ± 1.95 ^c^	6.98 ± 1.87 ^c^	7.18 ± 1.74 ^c^	7.16 ± 1.83 ^b^	6.51 ± 2.13 ^b^	6.93 ± 1.99 ^b^	6.84 ± 1.65 ^b^
TCCR	6.76 ± 1.91 ^c^	6.48 ± 1.88 ^d^	5.64 ± 2.22 ^c^	5.98 ± 2.02 ^d^	6.04 ± 2.02 ^d^	6.36 ± 1.49 ^d^	6.29 ± 1.38 ^c^	5.45 ± 1.83 ^c^	5.91 ± 1.77 ^c^	5.77 ± 1.86 ^c^
TCGH	5.52 ± 2.25 ^d^	4.61 ± 2.10 ^e^	3.96 ± 2.19 ^d^	4.74 ± 2.33 ^e^	4.38 ± 2.27 ^e^	5.16 ± 1.90 ^e^	4.48 ± 1.88 ^d^	4.59 ± 1.90 ^d^	4.48 ± 2.12 ^d^	4.34 ± 1.84 ^d^
**Soup with White Vegetables**										
	**Young**					**Seniors**				
	**Descriptors**									
**Kind of Edible Insects**	**Appearance**	**Odor**	**Taste**	**Structure**	**Acceptance**	**Appearance**	**Odor**	**Taste**	**Structure**	**Acceptance**
Control	9.06 ± 0.87 ^a^	8.48 ± 0.99 ^a^	8.36 ± 1.10 ^a^	8.48 ± 1.05 ^a^	8.60 ± 0.97 ^a^	9.07 ± 0.87 ^a^	8.45 ± 0.92 ^a^	8.47 ± 1.00 ^a^	8.18 ± 1.03 ^a^	8.42 ± 0.95 ^a^
WVMW	7.48 ± 2.03 ^b^	7.02 ± 1.79 ^b^	6.78 ± 2.03 ^b^	6.94 ± 2.03 ^b^	6.98 ± 1.74 ^b^	6.80 ± 1.96 ^b^	6.66 ± 1.59 ^b^	6.30 ± 1.95 ^b^	6.52 ± 1.95 ^b^	6.63 ± 1.70 ^b^
WVBW	7.22 ± 1.96 ^c^	6.96 ± 1.84 ^b^	6.78 ± 1.84 ^b^	7.02 ± 1.69 ^b^	6.86 ± 1.68 ^b^	6.86 ± 2.07 ^b^	6.64 ± 1.92 ^b^	6.40 ± 1.91 ^c^	6.71 ± 1.91 ^c^	6.59 ± 1.66 ^b^
WVCR	6.54 ± 1.96 ^d^	6.10 ± 2.07 ^c^	5.55 ± 2.41 ^c^	6.02 ± 1.95 ^c^	5.90 ± 2.03 ^c^	5.93 ± 1.61 ^c^	6.29 ± 1.49 ^c^	5.41 ± 1.92 ^d^	5.77 ± 1.96 ^d^	5.64 ± 1.87 ^c^
WVGH	4.68 ± 1.72 ^e^	4.47 ± 1.89 ^d^	3.72 ± 1.75 ^d^	4.09 ± 1.77 ^d^	4.31 ± 1.86 ^d^	4.98 ± 1.87 ^d^	4.78 ± 1.99 ^d^	3.55 ± 1.90 ^e^	4.96 ± 1.99 ^e^	4.23 ± 1.85 ^d^

A 10-point scale was used for the evaluation of the soups. The results are presented as mean and standard deviation. Values in the same column marked with different letters are statistically significantly different at *p* < 0.05.

**Table 3 nutrients-15-05047-t003:** Dependency of overall acceptance between the two age groups.

Kind of Soup	Acceptance	All	Young	Seniors	*p*
Tomato soup (STC)	Control	9.02 ± 0.91	9.04 ± 0.94	8.95 ± 0.75	0.359
	TCMW	7.10 ± 1.72	7.18 ± 1.52	6.98 ± 1.66	0.647
	TCBW	6.90 ± 1.87	6.98 ± 1.87	6.84 ± 1.65	0.682
	TCCR	5.92 ± 2.08	6.04 ± 2.02	5.77 ± 1.86	0.699
	TCGH	4.40 ± 2.11	4.38 ± 2.27	4.34 ± 1.84	0.839
White vegetable soup (SWV)	Control	8.61 ± 0.95	8.60 ± 0.97	8.42 ± 0.95	0.438
	WVMW	6.82 ± 1.80	6.98 ± 1.74	6.63 ± 1.70	0.296
	WVBW	6.73 ± 1.73	6.86 ± 1.68	6.59 ± 1.66	0.352
	WVCR	5.72 ± 2.34	5.90 ± 2.03	5.64 ± 1.87	0.527
	WVGH	4.27 ± 1.90	4.31 ± 1.86	4.23 ± 1.85	0.700

Comparison between two groups based on the Mann–Whitney U test. Value for *p* > 0.05.

**Table 4 nutrients-15-05047-t004:** Multiple equations of overall pancake acceptability.

Type of Additive	Regression Equation	R^2^
Control (SPC)	y = 0.31x_1_ + 0.05x_2_ + 0.29x_3_ + 0.35x_4_	0.99
*T. molitor* (TCMW)	y = 0.33x_1_ + 0.14x_2_ + 0.28x_3_ + 0.25x_4_	0.99
*A. diaperinus* (TCBW)	y = 0.19x_1_ + 0.16x_2_ + 0.31x_3_ + 0.34x_4_	0.99
*A. domesticus* (TCCR)	y = 0.15x_1_ − 0.05x_2_ + 0.59x_3_ + 0.31x_4_	0.98
*R. differens* (TCGH)	y = 0.07x_1_ + 0.06x_2_ + 0.61x_3_ + 0.26x_4_	0.97
Control (SWV)	y = 0.34x_1_ − 0.09x_2_ + 0.28x_3_ + 0.45x_4_	0.99
*T. molitor* (WVMW)	y = 0.26x_1_ + 0.16x_2_ + 0.22x_3_ + 0.35x_4_	0.99
*A. diaperinus* (WVBW)	y = 0.17x_1_ + 0.18x_2_ + 0.29x_3_ + 0.34x_4_	0.99
*A. domesticus* (WVCR)	y = 0.24x_1_ + 0.07x_2_ + 0.41x_3_ + 0.25x_4_	0.98
*R. differens* (WVGH)	y = 0.02x_1_ + 0.37x_2_ + 0.59x_3_ + 0.06x_4_	0.98

R^2^—determination coefficient, y—preference; x_1_— appearance; x_2_—odor; x_3_—taste; x_4_—texture.

## Data Availability

All data and material are included in the manuscript.

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
