# Peer review of "The Acceptance of Cream Soups with the Addition of Edible Insects (Mealworm, T. molitor; House Cricket, A. domesticus; Buffalo Worm, A. diaperinus; Grasshopper, R. differens) among Young People and Seniors in Poland"

_nutrients, 2023, doi:10.3390/nu15245047_

Round 1
Reviewer 1 Report
Comments and Suggestions for Authors
The authors present a paper on the acceptance of cream soups supplemented with edible insects flour. Paper writing and conceptualization appear adequate for the purpose of the journal. The topic is of interest for the scientific community and justifies the purpose of the research.
Introduction
The authors expose the state of the art of supplementation of food products with edible insect meals in a concise but effective form. However, it would be advisable to increase the bibliographical references on the acceptability by consumers of products supplemented with ingredients obtained from insects.
Materials and Methods
- Authors should provide (line 107) a detailed protocol or reference for the caloric value determination
M&M Sensory analysis section
- It is necessary to better clarify for what reason (line 136) some participants did not complete the study.
- The authors should justify the use of unstructured rating scales on non-trained judges, at least by stating that they are regular consumers of the product.
- The authors should provide maximum and minimum references ("edge labels") for descriptors used in sensory evaluation, or at least describe in detail the definition of descriptors.
Results
- The authors should better explain the reason for having age-extreme groups (line 208)
- There is a probable typo in line 226 (sponge cakes)
Overall conclusions
In the opinion of this reviewer, apart from the above-mentioned minor issues, the publication of the paper is worthy of consideration.
Author Response
Response do reviewer
Thank you very much for the review, the time spent, and the valuable suggestions, which I will try to apply and supplement the text.
Introduction
The authors expose the state of the art of supplementation of food products with edible insect meals in a concise but effective form. However, it would be advisable to increase the bibliographical references on the acceptability by consumers of products supplemented with ingredients obtained from insects.
In the introduction, valuable and current publications on consumer acceptability of products with added insects have been included.
Smarzyński, K.; Sarbak, P.; Musiał, S.; Jeżowski, P.; Piątek, M.; Kowalczewski, P.Ł. Nutritional analysis and evaluation of the consumer acceptance of pork pâté enriched with cricket powder - preliminary study. Open Agric. 2019,4(1),159-163.
Ors,i L.; Voege, L.L.; Stranieri, S. Eating edible insects as sustainable food? Exploring the determinants of consumer acceptance in Germany. Food Res. Int. 2019,125,108573.
Ribeiro, J.C.; Gonçalves, A.T.S.; Moura, A.P.; Varela, P.; Cunha, L.M. Insects as food and feed in Portugal and Norway–cross-cultural comparison of determinants of acceptance. Food Qual Prefer. 2022,102,104650.
Mishyna, M.; Chen, J.; Benjamin, O. Sensory attributes of edible insects and insect-based foods – Future outlooks for enhancing consumer appeal. Trends Food Sci. Technol. 2020,95,141-148.
Materials and Methods
- Authors should provide (line 107) a detailed protocol or reference for the caloric value determination
In line [112] information was added on which software was used to calculate the caloric value of a soup serving.
The caloric value was calculated using the 'Dieta 6.0' diet planning and energy value counting software, developed by the National Institute of Public Health, based on the Polish food composition database
M&M Sensory analysis section
- It is necessary to better clarify for what reason (line 136) some participants did not complete the study.
In line [136], information has been added regarding the reason why 16 young adults and 21 seniors did not complete the study. 21 seniors and 16 young adults did not complete the study. 4 seniors and 2 individuals from the younger group were ill or hospitalized, while the rest refused to continue the study or did not show up for the study without giving a reason.
- The authors should justify the use of unstructured rating scales on non-trained judges, at least by stating that they are regular consumers of the product.
Indeed, there was no information regarding the training of consumers for acceptance evaluation. The consumers were selected from the university's database and are not only regular consumers but also participated in previous similar analyses. Nevertheless, the person conducting the study explained the purpose of the research and the method of filling out the evaluation forms each time.
- The authors should provide maximum and minimum references ("edge labels") for descriptors used in sensory evaluation, or at least describe in detail the definition of descriptors.
The scale proposed for evaluation is a hedonic scale, on which the consumer only responds whether they like the product or not. The descriptor description was taken from the standard PN-EN ISO 5492:2009 PN-ISO 11035:1999, where there are no specific detailed edge descriptions.
Results
- The authors should better explain the reason for having age-extreme groups (line 208)
In line [214], the reason why extreme age groups were analyzed was further clarified. The study utilized two extreme age groups to indicate potential differences between the groups in the acceptance evaluation of insect-based products. The youngest adults are perceived as the most willing and open to trying new products and flavors. Meanwhile, individuals over the age of 65 are considered the most conservative group, reluctant to change dietary habits and try new items in the food market.
- There is a probable typo in line 226 (sponge cakes)
Thank you, the error has been corrected in line [226].
Reviewer 2 Report
Comments and Suggestions for Authors
It is a very relevant and important subject.
In table 1 the Latin names is not in italic. They should be.
What are the arguments for using tow decimal places in the tables?
In line 243 you write that tomato soup SCT was 9 and the cream white vegetable soup 8.5, but those numbers cannot be found in the table?
The text in the figures is far too small. It is too difficult to read.
Comments on the Quality of English LanguageIt looks as if the different authors have written different parts and not together which may be why there are used different wording. The word “organoleptic” is not correct to use, it should be sensory.
Author Response
Response to Reviewer
Thank you very much for the review, the time spent, and the valuable suggestions, which I will try to apply.
In table 1 the Latin names is not in italic. They should be.
In Table 1, the Latin names have been changed to italics.
What are the arguments for using tow decimal places in the tables?
Following the example of previous personal publications and those of other authors who published in Nutrients, it was decided to use two decimal places.
In line 243 you write that tomato soup SCT was 9 and the cream white vegetable soup 8.5, but those numbers cannot be found in the table?
Thank you for the valid remark. Indeed, the initial rating for SCT was 9.02 and for the vegetable soup 8.61, as presented in the table, but it was incorrectly recorded in the text. Changes have already been made in the manuscript text.
The text in the figures is far too small. It is too difficult to read.
The font in the legend and axis description has been changed in all charts. Currently, the charts are more readable.
It looks as if the different authors have written different parts and not together which may be why there are used different wording. The word “organoleptic” is not correct to use, it should be sensory.
The use of the term 'organoleptic' in this case was inappropriate, therefore it has been changed to 'sensory' throughout the manuscript, in accordance with the suggestion.